

# 1 Communicating the most accurate and reliable science on 2 climate change to society: A survey of editors from the 3 Intergovernmental Panel on Climate Change

Tomas Molina[1]& Ernest Abadal[2]
[1]Applied Phisics, Universitat de Barcelona
[2] Ernest Abadal, Universitat de Barcelona
Correspondence to: Tomas Molina (tomasmolinabosch@ub.edu)
**Abstract.** This study focuses on the perspectives of scientists involved in the IPCC AR5 and AR6 synthesis
reports, examining their views on the communication of climate change knowledge and its dissemination to the
public. The objectives include understanding scientists' opinions on the state of climate change knowledge, the
effectiveness of current communication strategies, and recommendations for improving public engagement. A
survey was conducted among 72 IPCC scientists, assessing their perceptions on various aspects of climate
communication, including the use of media, educational integration, and challenges like misinformation. Results
show that scientists generally rate the scientific community as well-informed, policymakers as moderately
informed, and the public as only acceptably informed about climate change. Many respondents suggested
improvements in the clarity and accessibility of IPCC reports, emphasizing the role of media, social networks,
and education in better informing the public. The study concludes that trust in information sources is vital for
effective climate communication and that a more tailored, empathetic, and solutions-based approach is
necessary to bridge the gap between scientific knowledge and public understanding.
Keywords: communication, climate, IPCC, survey, public

## 23 1 Introduction


The challenge posed by climate change to society is immense. The overwhelming evidence that human reliance
on fossil fuels has led to atmospheric warming, which in turn is altering weather patterns and the global climate,
highlights the need for widespread social awareness on a global scale. Few times in human history has there
been such an urgent need for a shared global consensus among all inhabitants of the planet (Somerville &
Hassol, 2011). Addressing and adapting to climate change requires not only agreement on a transition to new
energy paradigms but also discussions on the future of economic growth, or even potential degrowth (Hansen et
al., 2008; Howes et al., 2013). This consensus must be grounded in scientific knowledge, its credibility, and the
broad agreement within the scientific community (Buttel et al., 1990; Change, 2011; Fuhrer et al., n.d.).

The losses and damage already being caused by climate change, as well as those anticipated in the future,
highlight the fact that there will inevitably be both winners and losers in this global crisis. This reality extends



the discussion beyond the realm of science, touching on ethics, politics, ecology, sociology, and even religion
(Francisco, 2015). Addressing these multifaceted impacts requires an interdisciplinary approach that recognizes
the complex and far-reaching consequences of climate change on all aspects of society (Molina & Abadal,

39  2024).


In this highly complex context, science is expected to play a critical role in guiding decision-making and
shaping a unified global strategy for humanity's adaptation to these changes (Cutter et al., 2012). The
Intergovernmental Panel on Climate Change (IPCC) has emerged as the leading authority on expert knowledge
related to climate change. However, it is not without controversy (De Pryck, 2018). The influence of its
scientific reports on national and global policies often blurs the line between politics and epistemology, creating
tensions around the intersection of science and policy (Beck, 2012; Hermansen et al., 2021).

From its first report in 1990 to its sixth in 2023, the IPCC's level of certainty in its findings has steadily
increased. As a result, the urgency for action among decision-makers and society at large has intensified, giving
rise to terms like "climate emergency" and global agreements such as the Paris Agreement. The latter aims to
limit emissions and keep global temperature rise well below 2°C compared to pre-industrial levels (Höhne et al.,
2021; Molina & Abadal, 2021; Ripple et al., 2022).

The scientific foundation of the IPCC reports is derived from research published in peer-reviewed scientific
journals, which undergoes rigorous scrutiny by independent experts. Only knowledge that passes this
demanding review process is included in these reports. However, determining which findings are ultimately
incorporated into the reports that inform policymakers is itself a subject of analysis, attention, and, at times,
controversy (Beck & Mahony, 2018a).

Ultimately, the knowledge and strategies for mitigation and adaptation outlined in the IPCC reports are handed
over to policymakers, whose decisions impact society at large. The global strategy to combat and adapt to
climate change targets individuals across all social, cultural, and religious backgrounds, as well as those from
diverse economic and educational levels. The public's perception of the urgency, as well as the mitigation and
adaptation strategies outlined in the IPCC reports, extends beyond policymakers (Gemeda et al., 2023). These
reports form a key part of the information that reaches global society, which must ultimately support the
decisions made by political leaders. The popularization of the IPCC's findings—making complex scientific and
technical information accessible to the general public—requires an effective communication strategy. This
strategy should ensure that people of all knowledge levels can understand and engage with the content (Doran et
al., 2023; Rödder & Pavenstädt, 2023).

**2 Objectives**



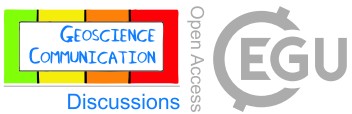

Our study group consists of scientists who were part of the writing teams for the IPCC5 and IPCC6 synthesis
reports. We are interested in their perspectives on the communication aspects of current climate change
knowledge, as well as their views on how effectively this information is being conveyed to the public.
The specific objectives are as follows:
1. To understand the perspectives of IPCC scientists on the current state of climate change and their role

78        in efforts to reduce and mitigate its impacts.

2. To gather opinions from IPCC scientists on how best to communicate the scientific content of IPCC

80        reports to the public.

3. To collect proposals from IPCC scientists on how to improve the dissemination of this scientific

82        information to society at large.

The scientific knowledge about climate change that reaches society must be both up-to-date and supported by
the broadest possible consensus within the scientific community. Additionally, this knowledge should be
presented in a way that is not only rational and easy to understand but also resonates with people on emotional
and spiritual levels across different cultures (Bolisani & Bratianu, 2018).
**3 Methodology**

The IPCC reports are published approximately every seven years, which can make it challenging to stay in
contact with the scientists who contributed to them. For this reason, we have focused our study on the two most
recent reports: IPCC AR6 and IPCC AR5. Our sample includes members of the Scientific Steering Committee
for the IPCC AR6 synthesis report (IPCC, n.d.), as well as the Chairs and Vice-Chairs of the IPCC AR5
synthesis report.

The fifth IPCC report was published in 2014, nine years before we launched our survey. As a result, some of the
scientists involved were no longer reachable at their original contact addresses. To address this, we searched
research publication databases for up-to-date contact information for both the IPCC AR5 and AR6 synthesis
report writing teams. After accounting for deceased individuals, we obtained a final sample of 28 contacts from
the IPCC AR6 and 44 from the IPCC AR5, resulting in a total of 72 contacts across the two reports.
The survey was structured into four sections: general information about the scientists, their perception of the
current level of knowledge on climate change, their views on the IPCC reports, and opinions on the
communication of these reports to society. We used closed-ended questions with a Likert scale, along with
open-ended options for questions related to communication.
The survey questions were reviewed by a scientist who contributed to both the IPCC AR6 and AR5 reports. To
rate the responses on the Likert scale, we assigned numerical values to each option, with 1 representing the
lowest value and 5 representing the highest. We then averaged the responses for each question or survey section.



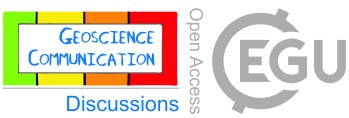

The resulting average reflects the respondents' positions based on the following scale: 1 = very low, 2 = low, 3 =
neither high nor low, 4 = high, 5 = very high.
To enhance the clarity of the results, we multiplied the average by two, converting the values to a scale of 1 to
10. The results were then classified using standard educational labels: "Very poor / F" from 0 to 2.9
"Insufficient / E from 3 to 4,9
"Sufficient / D" from 5 to 5,9
"Good / C" from 6 to 6,9
"Notable / B" from 7 to 8,9
"Excellent / A" from 9 to 10

The survey was distributed via email using a Google Forms format, with English as the language of
communication. It was initially sent out in February 2023, coinciding with the approval phase of the IPCC's
Sixth Assessment Report, which took place at the 58th panel session in Interlaken, Switzerland, in March of the
same year. A reminder was sent in April, after the approval process had been completed.

**4 Results and discussion**

The scientists who responded to the survey (figure 1) were aged 51 and older, with a significant portion (58.4%)
over 61. Although only one respondent explicitly identified as retired, the CVs of those who provided their
details indicate that some hold emeritus professor positions at their respective universities. The majority of
respondents were men (75%) and from academic institutions (83,3%). The representation of women, at 25% of
responses, aligns with the published gender demographics of IPCC report authors (Liverman et al., 2022). The
age distribution of our respondents is also consistent with findings from other studies on IPCC authors (Gay-
Antaki, 2021).

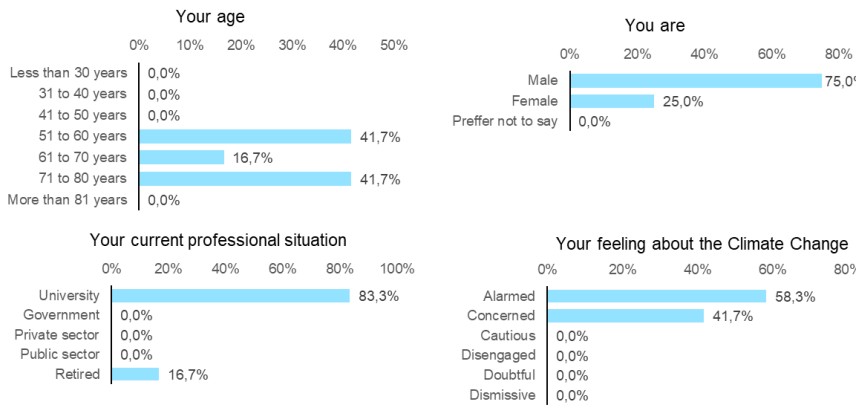


*Figure 1. Age, Gender, Profession & Feelings about Climate Change*

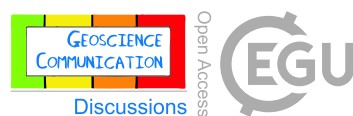


The majority of participants feel alarmed (58,3%) or worried (41,7%) about climate change. Those with a
deeper understanding of the current climate situation tend to view its potential future with greater concern and
alarm. This aligns with the evolution of the "Global Warming's Six Americas" framework, which illustrates a
growing concern about climate change and a shift in public attitudes over time (Leiserowitz et al., 2021).
Responses indicate that participants (figure 2) view the scientific community as highly informed about climate
change (rated 8,7) while they consider policymakers only moderately informed (rated between 6,3 (world) and
7,7 (local)). In contrast, the general public is seen as being only "acceptably" informed (rated between 6,2
(local) and 5,5 (world)). Participants also identified misinformation and biased, self-interested information as
notable issues (rated 8,3) The literature on climate change communication highlights several key points:
explaining its causes enhances science acceptance, emphasizing scientific consensus counters misinformation,
culturally aligned messaging is more effective, and inoculating against misinformation works best, though
debunking can also be successful.

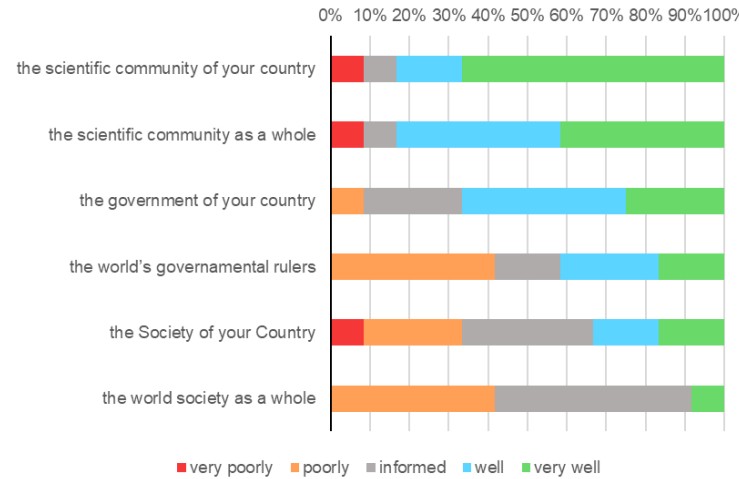

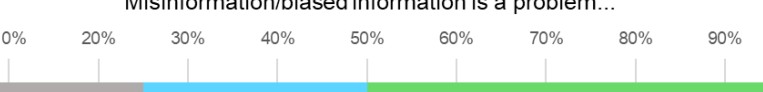


*Figure 2. Information Status, and Biased information*

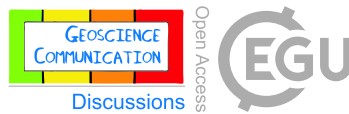

Regarding the IPCC reports (figure 3), the majority of participants believe they demonstrate notable scientific
objectivity (rated 8,8) and reflect the best available knowledge on climate change (rated 7,8). While respondents
feel that the reports have a notable impact on society as a whole (rated 7,5), opinions vary more widely in this
area.
The use of graphs and tables to enhance comprehension is highly appreciated (rated 9), as these visual aids make
the reports more understandable (Harold et al., 2020). Many respondents see the primary role of the IPCC
reports as providing the best possible information to decision-makers, rather than directly to the general public.
They believe that the public often accesses these reports through other interpretive channels, as expressed in
open-ended responses.

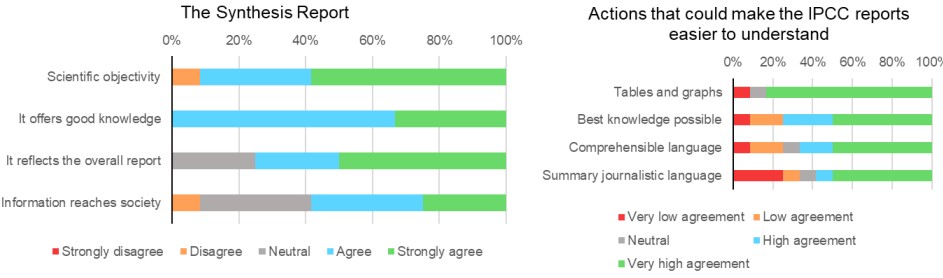


*Figure 3. About Summary Reports & Understanding of Reports*

Many respondents suggested the possibility of creating a more concise version of the IPCC summary report
specifically for the general public. Our survey findings align with the discussions and recommendations from
the IPCC's February 2016 Expert Meeting on Communications and their ongoing implementation. The goal is
to deepen understanding of the IPCC's communication efforts within the broader context of climate
communication and policy. This may also inspire further ideas on how to strengthen the IPCC's communication
strategies (Lynn, 2018).
When discussing how to communicate the contents of IPCC reports to the public, the majority (rated 9,2)
believe it is appropriate for these reports to be integrated into university curricula and school education (rated
9,2). The strong agreement among our survey respondents aligns with literature indicating that university
students believe climate change is real and primarily human-induced, with the majority expressing concern.
However, the studies also reveal misconceptions about the fundamental causes and consequences of climate
change (Wachholz et al., 2014).
Respondents also emphasized the importance of making the reports fully accessible to everyone via the internet
(rated 9). Social networks (rated 9,3), along with media outlets like television (rated 9) and radio (rated 9), were
seen as the most suitable platforms for informing the public. The written press was rated slightly lower (rated
8.8), but still viewed as an important channel. Overall, respondents rated highly the effectiveness of these

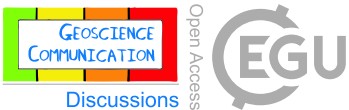

channels for informing the population. The use of new media aligns with studies suggesting that non-elite
actors, such as individual bloggers and concerned citizens, are effective climate change advocates. While
mainstream media remains the most frequently discussed, new media and science information sources are strong
competitors for audience attention (Newman, 2017).
It is worth noting that some respondents expressed dissenting opinions on the use of journalistic language,
political debates, and religious or spiritual sermons in the communication of these reports. The simplification of
scientific information often risks undermining its credibility, largely due to a failure to recognize the tensions
between scientific and public interpretations. Maintaining scientific credibility requires balancing it with
meaningful social and political dialogue about the values we hold and the actions we take to protect them.
Strengthening the link between the theory and practice of climate science communication is essential (Hollin &
Pearce, 2015; Pidcock et al., 2021).

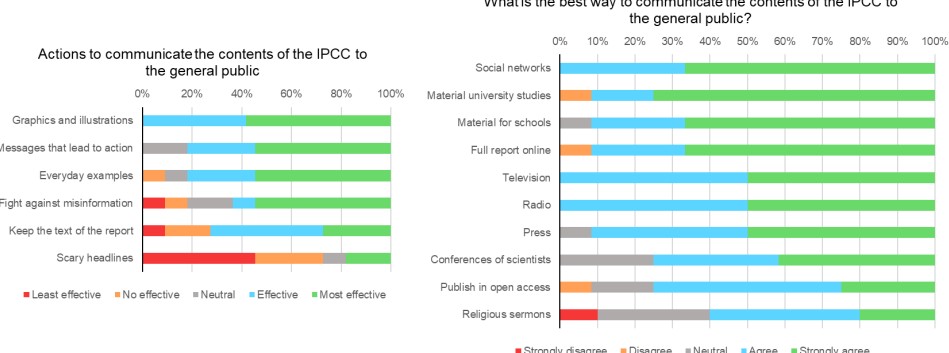


*Figure 4. Public Communication and Channels*

Misinformation is widely seen as a significant problem (rated 7,8). This issue was raised three times throughout
the survey (Sanford et al., 2021), and in both instances where respondents were asked whether misinformation
was a concern, the responses were remarkably consistent. There was even stronger agreement on the need to
actively combat misinformation (Lewandowsky, 2021).
According to respondents, the biggest challenge in communicating climate change is not the difficulty of
understanding its scientific aspects (rated 6,5), nor simply the need to convey clear and relevant information to
users (Adler & Hirsch Hadorn, 2014). Instead, the primary challenge lies in the complexity of decision-making
within social and economic contexts (rated 8,3). As highlighted in the literature, this complexity reflects an
evolving relationship between climate science and policy, which is undergoing a significant transformation
(Beck & Mahony, 2018b).
Additionally, the vast majority of participants provided comments and suggestions in the open-ended questions.
Among the most commonly suggested solutions were:





• Short, simple, and easy-to-understand messages, that may help in making IPCC a power

204       communicating tool (Stocker & Plattner, 2016).

• Demonstrating empathy towards individuals and communities by linking climate change to everyday

206       life and focusing on the future of new generations, while staying true to the content of the reports

207       (McBeth et al., 2022).

One notable response from Ethiopia highlighted the need to improve the training of those responsible for
informing the public about climate change.

**Conclusions**

Participation in our survey was relatively low, with only 16.6% of the sample responding. The lack of
engagement from key IPCC scientists, who are responsible for preparing the institution's most widely read
reports, aligns with findings from previous studies highlighting the difficulty that the average reader has in
comprehending these reports (Dormer, 2020; Jos Delbeke et al., 2019).
One lead author of IPCC AR6 WGII declined to participate in the survey because it did not allow the option to
leave questions blank or to skip options, they felt were insufficiently detailed. Another scientist, a vice-chair of
the IPCC AR6, completed the survey but expressed, both in the open-response section and via email, their
disagreement with several concepts and requested that some of his responses be disregarded.
The disparity in participant behavior in our survey is evident: while some respondents scored highly on
questions involving concepts such as journalistic language, religion, or politics, others either refused to
participate or expressed dissatisfaction with the inclusion of these topics. This highlights the need for a revised
communication strategy that addresses these concerns and enhances the impact of the IPCC report content
(Anseel et al., 2010; Bhandari, 2022; Solecki et al., 2024).
Trust in the source of information is crucial for that information to influence decision-making. This relationship
between trust and decision-making has been extensively studied in medicine, particularly in managing the
delivery of "bad news" and the need for patients to make significant decisions. Informed decision-making is
now a well-established practice in medical fields (Chandra et al., 2018; Musmade et al., 2013).
Information about climate change often represents "bad news" for much of society, requiring careful
communication and informed decision-making. Trust in the source of climate information is just as essential as
it is in medicine. Similar to medical contexts, recipients of climate information often lack the full capacity to
understand highly technical or scientific content, especially during times of emotional stress. Therefore, this
information needs to be adapted to the audience's level of understanding. The scientists' responses in our survey,
which emphasize the need for empathy and a solutions-based approach, reflect this mindset. While the diagnosis
and proposed solutions must come from science, their implementation requires clear communication to society,



which must ultimately make the final decision—ideally, with widespread social consensus (Goldberg et al.,
238 2020).

The open-ended responses to our survey also highlight the critical role of trust in the information source
(Goodwin & Dahlstrom, 2014). National Meteorological Services serve as key guarantors of the accuracy and
reliability of past climate data, which underpins their credibility when comparing past and present data to
confirm that climate change is occurring. They also play a vital role in explaining the new climate realities to
society, allowing people to comprehend and contextualize the future climatology they will face (Molina &
Abadal, 2024).
Communication is a broad concept that encompasses the sender, the receiver, and the message. However, it also
involves the action (or inaction) of communicating, beyond the mere intentions of those sending and receiving
information (Charles Bazerman, 2019; Luhmann, 1992). In the case of climate change, where the active
participation of the public is crucial, a proliferation of diverse and engaging narratives around the topic is
necessary to inspire action and understanding.
These narratives must be tailored to the diverse cultures, beliefs, and values of different human communities
worldwide, offering a moral framework that is acceptable to all (Hulme, 2009). Climate change communication
involves many stakeholders with varying levels of expertise and perspectives, yet all rely on the scientific
foundation of climate knowledge. How this knowledge reaches and resonates with society is crucial, and the
process of popularizing it should involve the scientists who created it. Developing a methodology within the
IPCC itself to produce texts written in clear, accessible language—akin to journalistic style (Smith & Higgins,
2020) —could help reduce the contradictory and confusing headlines that often reach the public. Some IPCC
scientists who responded to our survey suggested that this could be an innovation for future cycles, proposing
ideas such as creating a summary text for the general public, approved by scientists rather than governments, to
eliminate concerns about politicization and preserve trust in both the message and its source.
The role of the media and social networks in interpreting and delivering IPCC information to the public is vital,
as reflected in the opinions of our respondents. Media coverage and social media discussions shape public
opinion on climate change (Pearce et al., 2019; Sarathchandra & Haltinner, 2023). The media's portrayal of the
Conferences of the Parties (COP), where decision-makers, government representatives, and non-governmental
organizations gather, also influences societal perceptions of climate action and the acceptance of measures to
mitigate climate change, whether current or forthcoming (Sisco et al., 2021).
In recent years, significant research has explored the role of emotions, empathy, and affect (Brosch, 2021) in
climate change communication, aiming to inspire societal action. However, the gap between climate scientists
and the delivery of their findings to the global public remains unresolved. More efforts are needed to create
content that can be directly communicated to society without the often inaccurate interpretations introduced by
intermediaries who currently serve as the public's source of climate information.



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

I have collaborated with assessment agencies in Catalonia (AQU, AGAUR), Spain (AEI, ANEP, ANECA,
ACSUCYL), Italy (ANVUR) and Greece (HQA).



**-Ethics Approval**
There are no conflicts of interest among the authors, and no external funding was involved in this research. All
participants were fully informed about the purpose of the study and provided their consent to participate in the
survey.

**-Consent for publication**
All authors consent to participate.



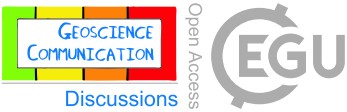


**-Competing Interests**

The authors have no relevant financial or non-financial interests to disclose.


**-Author contributions (Please ensure that all authors are individually mentioned in the author contribution statement.)**

TM, EA design, conceptualization

TM, EA data acquisition

TM, EA analysis and data interpretation

TM Article Writing

TM, EA article review

All authors read and approved the final manuscript.

493

**-Funding**

The authors declare that no funds, grants, or other support were received during the preparation of this manuscript.

497

**-Availability of data and materials**

Survey and survey results included in Supplementary Materials

500