# Peer review of "Communicating the most accurate and reliable science on"

_Geoscience Communication, 2024_

## Referee Comment (RC3)

**Communicating the most accurate and reliable science on climate change to society: A survey of editors from the Intergovernmental Panel on Climate Change**
Author(s): Tomas Molina and Ernest Abadal
MS No.: gc-2024-8
MS type: Research article

**1. General comments**

Thank you very much for the interesting and valuable propostion to survey the IPCC scientists on communicating climate science.

I have taken into consideration the annexed "Questionaire" (survey questions) that the authors kindly provided in answer to Referee #1 (RC1).

I am concerned, however, that the survey questions are vague and there is not enough information on the respondents and their communication expertise to understand if they answered the survey (assesing, for example, effectiveness of communication methds) based on their past experiences or if their answers were speculative opinions based on their intuition. While the latter one is interesting to read and answers to one of the study's objectives, "understanding scientists' opinions", only the former one could empirically serve the study's aim to form "recommendations for improving public engagement".

This aligns to lines 235-236 ("While the diagnosis and proposed solutions must come from science, their implementation requires clear communication to society"): the science expertise is one phase, the effective communication is another. It is not clear if the results presented here could be applied to form a clear communication strategy.

I would kindly suggest to the authors to include a richer and more up-to-date source of references to back up several statements throughout the manuscript. I have made specific suggestions below.

I would also kindly suggest to the authors to address the limitations of the survey more clearly in the Discussion section, including the rather homogenous pool of respondents (male, senior, university, as well as 0% work in government, public or private sector).

I would recommend for future iterations, to offer a more detailed profile of the surveyed scientists: the degree to which they have previously been involved in communicating (climate) science to non-scientific audiences, as well as which research fields they belong to, for example. Currently knowing that 83% work in university gives us only a homogenous profile. Yet reports on the use of social media indicate that their use is very demographic dependent. Knowing more about the experience of the respondents with communication tools or their profile may be relevant to the survey questions asking for their opinion regarding the best actions/platforms to communicate climate science (Figs 3-4).

**2. Specific comments**

The introduction would benefit from additional up-to-date references.

Line 14:
"Results show that scientists generally rate the scientific community as well-informed, policymakers as moderately informed, and the public as only acceptably informed about climate change."
1. Which criteria did scientists base the degree of "informed-ness", and how did they assess these criteria across stakeholders: eg, policymakers versus public?

Line 25
"The challenge posed by climate change to society is immense."
2. The references in this paragraph are rather old, from 1990-2013. I would recommend complementing them with newer citations.

Line 25
"The overwhelming evidence that human reliance on fossil fuels has led to atmospheric warming, which in turn is altering weather patterns and the global climate, highlights the need for widespread social awareness on a global scale."
3. Please include references to some of that evidence, perhaps literature reviews. I suggest it may be beneficial to include among the earliest references that find evidence of the warming effect of fossil fuels to emphasise the need for awareness you mention.

Line 28:
"Few times in human history has there been such an urgent need for a shared global consensus among all inhabitants of the planet (Somerville & Hassol, 2011)."
4. Your single reference is from 2011. I would suggest adding a reference to an example from covid that induced an urgent, global evidence-based response and was dependant on communication to the general public.

Line 29:
"Addressing and adapting to climate change requires not only agreement on a transition to new energy paradigms but also discussions on the future of economic growth, or even potential degrowth (Hansen et al., 2008; Howes et al., 2013)."
5. Again, I think this statement would also benefit from including up-to-date references. Perhaps it may also be relevant to add a reference to the IPCC adaptation and mitigation report.

Lines 34-35:
"The losses and damage already being caused by climate change, as well as those anticipated in the future, highlight the fact that there will inevitably be both winners and losers in this global crisis"

6. References are missing for this statement.

Line 41:
"In this highly complex context, science is expected to play a critical role in guiding decision-making and shaping a unified global strategy for humanity's adaptation to these changes (Cutter et al., 2012)"

7. I would recommend including a reference to research that highlights the importance of using scientific evidence to guide decision-making in the context of climate action.

Line 60:
"Ultimately, the knowledge and strategies for mitigation and adaptation outlined in the IPCC reports are handed over to policymakers, whose decisions impact society at large"

8. It would be interesting to include references to a study of the impact of IPCC reports on policymaking, such as the Paris Agreement.

Line 65:
"reports form a key part of the information that reaches global society, which must ultimately support the decisions made by political leaders."

9. The order of the words implies the reports should support the decisions of political leaders; do you rather mean, the political leaders should uptake these reports in their decision making?

Line 67:
This strategy should ensure that people of all knowledge levels can understand and engage with the content (Doran et al., 2023; Rödder & Pavenstädt, 2023)."

10. To conclude the introduction section, I would suggest adding a sentence stating the aim of this paper, i.e. how this study aims to contribute to such a communication strategy.

Line 84:
"Additionally, this knowledge should be presented in a way that is not only rational and easy to understand but also resonates with people on emotional and spiritual levels across different cultures (Bolisani & Bratianu, 2018)."

11. I would recommend adding more studies on the effect of empathy, psychology (behavioural science), and overall emotional connection for effective climate communication. Perhaps include how on the other hand, there is a negative effect to contend with, such as climate anxiety.

**Methodology**

The methodology could include more details on the design of the questions, the review process of the questions, and how the open-ended questions were analysed. Below please find specific comments:

Line 100:
"The survey was structured into four sections: general information about the scientists, their perception of the current level of knowledge on climate change, their views on the IPCC reports, and opinions on the communication of these reports to society."
    12. How were the survey questions designed and by whom? Was a survey expert consulted?

Line 103:
"open-ended options for questions related to communication"
    13. How were open ended answered analysed? There is little mention of the outcome of this section in the Discussion. Could the answers be anonymized and shared in the annex?

Line 104:
"The survey questions were reviewed by a scientist who contributed to both the IPCC AR6 and AR5 reports. "

    14. Was there a testing phase to assess clarity/ambiguity of the questions before the final survey was sent out?
    15. How was this one scientist chosen to be the reviewer and what they reviewed (clarity of questions? Appropriateness of language?), or if the had expertise in survey assessment/design.

**Results and discussion**

Line 154-155:
"Many respondents see the primary role of the IPCC reports as providing the best possible information to decision-makers, rather than directly to the general public."
    16. Is the statement coming from the open-ended questions?

Line 170:
"students believe climate change is real and primarily human-induced.."
    17. I would suggest to exchange the word 'believe' to 'understand' or similar. In my opinion, it doesn't serve the scientific community to use the word 'believe' in the climate discussion.

Lines 173-182:

18. This paragraph describes the results of the survey where social media is rated as a most efficient platform to communicate, with religious sermons in the other extreme. I would respectfully argue that not knowing what the respondents base this assessment on (personal experience with social media? past collaboration with or presentations during a religious sermon?), it is hard to read an opinion on how 'effective' a platform is.

Lines 183-185:
19. These statements could be supported by references to studies on the effect of scientific language, values, etc on scientific credibility.

Lines 202-207:
20. You are listing examples of the most common open-ended questions. It is not then clear why there are references to the two listed proposals. Did participants all coincide with these references? Or are you adding supporting references to their opinions?

Line 208:
21. You make no prior reference to the geographical representation of the respondents. Including 'Ethiopia' here seems needless, unless their suggestion is particularly relevant coming from that country. I suggest rephrasing it to 'one notable response included...".

**Conclusion**

Line 214-2016:
22. It is not clear to me the relation between high-profile IPCC authors responding the survey and the ability of the average reader to understand the report. Please clarify, if you agree.

Line 223-224:
"...expressed dissatisfaction with the inclusion of these topics. This highlights the need for a revised communication strategy that addresses these concerns and enhances the impact of the IPCC report content"
23. It is not clear from this sentence if respondents expressed dissatisfaction with having to answer questions on these topics in this survey, or in the inclusion of these topics in the discussion of climate communication in general. Your call to action to address these concerns would suggest the latter, but it is not clear to me from this sentence which it is. You had mentioned in line 217-218 that one of the respondents, lead author of the IPCC, declined to participate in the survey due to the questions of the survey.

Line 255:
24. You suggest using "clear, accessible language—akin to journalistic style" for the general public; however over 30% of your respondents (figure 3) seem to have

"low-" or "very low agreement" that such an action would make the report easier to understand. Could you please add a comment on this to the discussion?

Line 267-268:

25. When discussing a gap in effective communication between scientists and the public, I would suggest bringing to the discussion the concepts of knowledge brokers or boundary organisations, which is not discussed at all in this study. For example, a call for knowledge brokers and boundary organisations – albeit in the context of evidence-informed policy making- is the European Commission's JRC "Commission staff Working Document" (SWD(2022) 346).

**3. Technical corrections**

26. Some decimal points are marked by periods (full stops) and some by commas. Please make it consistent. eg.  Line 123-126: "The scientists who responded to the survey (figure 1) were aged 51 and older, with a significant portion (**58.4%**) over 61 [...] The majority of respondents were men (75%) and from academic institutions (**83,3%**)"

Line 132 -

27. Figures captions don't need to capitalise each word, eg in: "Age, Gender, Profession & Feelings about Climate Change"

Line 134-135:
"Those with a deeper understanding of the current climate situation tend to view its potential future with greater concern and alarm."

28. I understand you are referring to the surveyed IPCC scientists versus the general public in this line. It may be helpful to clarify you are comparing the respondents to the general public to avoid confusion since the previous line is comparing the different levels of worry of only the surveyed scientists.

Line 159- Figure 3 caption

29.  You use the word "Summary" report, and in the figure it is 'Synthesis report'. Please make consistent.

30. The phrase 'summary journalistic language' is not proper English. Please amend.

Line 189:

31. Figure 4: The figure legend has a typo: "No effective" (green). Change to 'Not effective'.

Line 216:

32. Order of reference out of chronological order.

**Supplement**

The supplement pdf seems to be have truncated questions. Kindly verify the file is correct.

---

## Author Comment (AC1)

Major comments

Thank you for including a breakdown of the age, gender and profession of the IPCC scientists you interviewed. I think data on their nationality is also needed – maybe broken down by world region?

Many thanks for your comments. We are sorry but we did not include a question about nationality in our survey. Our point for not doing so is that IPCC rules are to be balanced in terms of nationalities of the scientific teams, and our research focus is mainly in the perspectives of the communication to the public of the IPCC findings.

I recommend including a copy of the full questionnaire, perhaps in an appendix. I think this is always good practice for any study about a survey. However, I think it is particularly important in your study, because you mention in the conclusion that some scientists had some concerns with parts of the questionnaire.

Many thanks for this suggestion. We did submit the questions and answers data, but we have also added a file with the full questionnaire.

Figure headings need to be clearer  – it might work well if you use questions in the survey as your figure headings?

For example, figure 4b has the headline "What is the best way to communicate the contents of the IPCC to the general public?" This is a good headline, as is explains the responses.

However, figure 3a has a headline "The synthesis report". This does not tell the reader what question the scientists are answering.

Many thanks for your points that help us to clarify our text. As you will see in the full questionnaire, this question was to place e statements about the IPCC text. We have changed the first statement to clarify the answers.

Figure 2b has the heading "misinformation/biased information is a problem", with answers ranging from "very important" to "not important at all". Grammatically, the answers do not seem to match the question.

Many thanks for your comments. We think that in the context of the survey the, the phrases proposed are understandable and do not lead to misunderstanding on the answers.

Misinformation/biased information is a problem… Not important at all

Misinformation/biased information is a problem…Somewhat important

…

Misinformation/biased information is a problem…Very important

Minor comments

There are instances where the text of your article does not match the figures.

Line 141 refers to "misinformation and biased, self-interested information", whereas the figure refers to "misinformation/biased information".

Many thanks for your point. We have amended it in text.

Line 134 uses the words "alarmed" and "worried", whereas the figure uses "alarmed" and "concerned".

Many thanks for your point. We have amended it in text.

The findings from your study sometimes mix with your citations in a confusing way – for example in lines 191-200. For clarity, the text could be amended to make a clearer distinction between your work and other peoples' work

We have amended the text following your kind suggestion

In figure 3b, "Actions that could make IPCC reports easier to understand", I am not sure what the "best knowledge possible" option means? My understanding is that the IPCC already uses the best body of scientific knowledge available. Please correct me if I am misunderstanding the question or answer, and/or clarify the text to make it clearer.

Many thanks for your comment. The literal text of the proposed answer is: They must concentrate on embodying scientific knowledge on each subject (the question is: 7 IPCC reports tend to be quite technical and extensive, and not always easy to understand. What are the features that can facilitate comprehension to people outside the field of science, or with less training?)

We have changed the text of the figure  to  Embodying scientific knowledge

I don't understand the conclusion drawn in lines 213-216. Are you arguing that the lack of engagement by the IPCC linked to the fact that the average reader has difficulty understanding the reports.

This is exactly our conclusion!

In fact this paper is part of one of the authors Phd about "Climate change communication: actions and strategies to increase public awareness and improve decision-making" that will be defended in the third week of january 2025.

Do you have any recommendations for the experts working on AR7 based? I would be interested to hear them - maybe in a discussion section?

Surely we have!
As a result of our research, and in form of a corollary of the Phd,we presented a communication at the EMS conference in Barcelona
https://meetingorganizer.copernicus.org/EMS2024/EMS2024-211.html
This research has been sent to be published as a letter to the editor.

---

## Author Comment (AC2)

**IPCC QUESTIONNAIRE**

Objective: To know the opinion of IPCC Synthesis Report Chairs & Vice-chairs, on aspects related to the communication of climate change to society.
This is a research that is part of a doctoral thesis
Approximate completion time: 10 minutes.
Thank you very much for your collaboration
Tomas Molina
University of Barcelona

- The data collected complies with the European "General Data Protection Regulation" and will be used exclusively for research purposes.

**You authorize your answers to be used for a scientific article**
Yes
No

**A Personal information**

**1 Your age ***
less than 30 years
31 to 40 years
41 to 50 years
51 to 60 years
61 to 70 years
71 to 80 years
more than 81 years

**2 You are ***

male
female
preffer not to say

**3 Please choose the one that better fits your principal current professional situation ***

University
Government
Private sector
Public sector
Retired
Other…

**B On the state of knowledge about Climate Change**

**4 Which one from this list, better describes your feeling about the Climate Change ***

Alarmed
Concerned
Cautious
Disengaged
Doubtful
Dismissive

**5 How informed is about climate change...***
(1: very poorly informed / 5: very well informed)

the Society of your Country
the government of your country
the scientific community of your country
the world society as a whole
the world's governamental rulers
the scientific community as a whole

**6 Disinformation or biased information against Climate Change poses a problem … ***
(Rate from 1 Not important at all and 4 very important)

**C On the IPCC reports**

**7 IPCC reports tend to be quite technical and extensive, and not always easy to understand. What are the features that can facilitate comprehension to people outside the field of science, or with less training? ***
( 1: strongly disagree / 5: strongly agree )

They must be written in language that is as much understandable as possible
They must concentrate on embodying scientific knowledge on each subject
They must include resources such as graphs or tables to facilitate understanding.
They must contain a summary expressed in journalistic language so that they can be understood by the general public

**7b Other features that can facilitate comprehension to people outside the field of science, or with less training.**
…

**8 The synthesis report that is approved by the government representatives is perhaps the report with the most informative relevance and that generates the most media attention. ***
(1: strongly disagree / 5: strongly agree)

The synthesis report reflects the content of the corresponding full report
The approval procedure maintains the perception of the scientific objectivity of knowledge
From reading the synthesis report, you can get an adequate general knowledge of the situation
The interpretation that comes to society about this report is appropriate

**D On the importance of communication in the IPCC**

**9 The fact that the IPCC reports are the main scientific reference on Climate Change…. ***
(1: strongly disagree / 5: strongly agree)

means that there are limitations in the communication of scientific outcomes to the general public.
forces to maintain the original text of the reports in all communications.
limits the possibilities of making headlines.
limits the possibility of creating simplified texts for the majority of audiences.

**10 Which actions do you consider most appropriate (effective) to communicate the contents of the IPCC reports to the general public ***
(1: least effective / 5: most effective)

Give examples from everyday life
Stick to the original text of the reports
Make headlines that frighten the public
Use graphics and illustrations
Extract messages that lead to action
The fight against scientific misinformation

**10b Other actions to communicate IPCC contents**
…

**11 What is the best way to communicate the contents of the IPCC to the general public? ***
(1: strongly disagree / 5: strongly agree)

Publish the complete IPCC reports on the Web
Open access of all research publications on Climate Change
Try to make the IPCC reports part of the training of university students
Ensure that the core messages of the IPCC reports become part of the schooling curriculum
Written press information
Information on televisions
Information on the radio
Diffusion on social networks
Conferences of scientists
Sermon of spiritual and religious leaders

**12 The main problem of Climate Change communication is…***
(1: strongly disagree / 5: strongly agree)

the scientific complexity of the subject.
the uncertainty in the scientific statements.
the social and economic complexity in decision-making.
the political controversy.
the self-interested misinformation.

**13 As a scientist, what would you emphasise about communicating the science of Climate Change to the general population**

**E Results**
If you want to receive the results of the questionnaire, please write your email

---

## Author Comment (AC4)

REPLY 2 RC1

Hi,

Thank you for addressing my comments so thoroughly!

There is only one point that I would like to reiterate – I think data on which country IPCC authors are from would be helpful. The questionnaire specifically asks IPCC authors about the scientific community/government in their country, e.g. in Figure 2, so I think this information would be useful.

I know that that the IPCC tries to achieve a global balance in their authors, but analysis shows that this has not been achieved. E.g. https://www.carbonbrief.org/analysis-how-the-diversity-of-ipcc-authors-has-changed-over-three-decades/.

Even if you did not ask IPCC authors for their nationality for the survey, it is possible to find which country IPCC scientists are based in here https://archive.ipcc.ch/report/ar5/authors.php

Many thanks for your kind suggestion that helps us to improve our paper. We have added results about nationality from those respondents who identified themselves at the survey. The text is as follows:

Out of all respondents, six chose to identify themselves to receive the survey results. Four were from Europe (the UK, France, the Netherlands, and Switzerland), while two were from Africa, both hailing from Ethiopia but affiliated with different IPCC cycles.

Of course, this is just a suggestion. But I feel that it would be a useful addition to your study.

---

## Author Comment (AC6)

REPLY RC3

Communicating the most accurate and reliable science on climate change to society: A survey of editors from the Intergovernmental Panel on Climate Change Author(s): Tomas Molina and Ernest Abadal MS No.: gc-2024-8 MS type: Research article

 1. General comments

Thank you very much for the interesting and valuable propostion to survey the IPCC scientists on communicating climate science.

 I have taken into consideration the annexed "Questionaire" (survey questions) that the authors kindly provided in answer to Referee #1 (RC1).

I am concerned, however, that the survey questions are vague and there is not enough information on the respondents and their communication expertise to understand if they answered the survey (assesing, for example, effectiveness of communication methds) based on their past experiences or if their answers were speculative opinions based on their intuition. While the latter one is interesting to read and answers to one of the study's objectives, "understanding scientists' opinions", only the former one could empirically serve the study's aim to form "recommendations for improving public engagement".

This aligns to lines 235-236 ("While the diagnosis and proposed solutions must come from science, their implementation requires clear communication to society"): the science expertise is one phase, the effective communication is another. It is not clear if the results presented here could be applied to form a clear communication strategy.

I would kindly suggest to the authors to include a richer and more up-to-date source of references to back up several statements throughout the manuscript. I have made specific suggestions below.

I would also kindly suggest to the authors to address the limitations of the survey more clearly in the Discussion section, including the rather homogenous pool of respondents (male, senior, university, as well as 0% work in government, public or private sector).

I would recommend for future iterations, to offer a more detailed profile of the surveyed scientists: the degree to which they have previously been involved in communicating (climate) science to non-scientific audiences, as well as which research fields they belong to, for example. Currently knowing that 83% work in university gives us only a homogenous profile. Yet reports on the use of social media indicate that their use is very demographic dependent. Knowing more about the experience of the respondents with communication

tools or their profile may be relevant to the survey questions asking for their opinion regarding the best actions/platforms to communicate climate science (Figs 3-4).

**GENERAL REPLY**:

Thank you for your thoughtful and constructive feedback, which will undoubtedly help improve our paper. Below, we address your comments in detail.

**Survey Scope and Aim:**
As outlined in our manuscript, the primary aim of this research is to explore the perspectives of scientists involved in the IPCC AR5 and AR6 synthesis reports regarding the communication of climate change knowledge to the public. Specifically, we sought to understand their opinions on how IPCC messages could be more effectively disseminated. We deliberately targeted editors of the IPCC Summary for Policymakers, as they play a central role in shaping the communication of this vital information, regardless of their formal communication training.

**Regarding Communication Expertise:**
We appreciate your concern about the respondents' communication experience and its influence on the validity of their responses. Our study prioritizes capturing their perspectives as scientists and coordinators of the IPCC synthesis, recognizing that they bring a unique vantage point to this issue. However, we acknowledge that their views may reflect both their professional expertise and personal intuition. We will revise the manuscript to clarify this distinction and its implications for interpreting the results, as well as to better situate our findings within the broader context of science communication research.

**Survey Limitations:**
We agree that the homogeneity of the respondent pool is a limitation, as noted in your feedback. While this cohort represents a highly influential group within the climate science community, their perspectives may not fully encompass the diversity of views or experiences in science communication. We will address this limitation more explicitly in the Discussion section, acknowledging that the lack of demographic and professional diversity among respondents could affect the generalizability of our findings.

**Future Research Directions:**
Your suggestion to include a more detailed respondent profile in future surveys is well-taken. Capturing information about their prior communication experience, research fields, and demographic characteristics would provide richer context for interpreting their responses. Additionally, as you noted, understanding the role of demographic factors in social media use could enhance our insights into effective communication strategies. We will incorporate this recommendation into the Discussion section as a valuable avenue for future work.

**Additional References:**
We will also review and incorporate more recent and relevant references to strengthen the manuscript and support key statements, as per your suggestion.

Once again, we appreciate your detailed feedback and are confident that addressing these points will improve the rigor and impact of our study.

2. Specific comments

The introduction would benefit from additional up-to-date references.

Many thanks for your point, we have added references from year 2020
Line 14:

"Results show that scientists generally rate the scientific community as well-informed, policymakers as moderately informed, and the public as only acceptably informed about climate change."

1. Which criteria did scientists base the degree of "informed-ness", and how did they assess these criteria across stakeholders: eg, policymakers versus public?

Many thanks for your comment. This paragraph is the description of the replies from the opinion of the editors of the summary for policymakers of the IPCC Assessment reports 5 and 6

Line 25 "The challenge posed by climate change to society is immense."

2. The references in this paragraph are rather old, from 1990-2013. I would recommend complementing them with newer citations.

Many thanks, we have added citations up from 2020

Line 25 "The overwhelming evidence that human reliance on fossil fuels has led to atmospheric warming, which in turn is altering weather patterns and the global climate, highlights the need for widespread social awareness on a global scale."

3. Please include references to some of that evidence, perhaps literature reviews. I suggest it may be beneficial to include among the earliest references that find evidence of the warming effect of fossil fuels to emphasise the need for awareness you mention.

Many thanks, we have added the reference to the IPCC 6 Climate Change 2021 The Physical Science Basis

Line 28: "Few times in human history has there been such an urgent need for a shared global consensus among all inhabitants of the planet (Somerville & Hassol, 2011)."

4. Your single reference is from 2011. I would suggest adding a reference to an example from covid that induced an urgent, global evidence-based response and was dependant on communication to the general public.
Many thanks, we have added citations up from 2020

Line 29: "Addressing and adapting to climate change requires not only agreement on a transition to new energy paradigms but also discussions on the future of economic growth, or even potential degrowth (Hansen et al., 2008; Howes et al., 2013)."

5. Again, I think this statement would also benefit from including up-to-date references. Perhaps it may also be relevant to add a reference to the IPCC adaptation and mitigation report.

Many thanks, we have added citations up from 2020

Lines 34-35: "The losses and damage already being caused by climate change, as well as those anticipated in the future, highlight the fact that there will inevitably be both winners and losers in this global crisis"

6. References are missing for this statement.

Line 41: "In this highly complex context, science is expected to play a critical role in guiding decision-making and shaping a unified global strategy for humanity's adaptation to these changes (Cutter et al., 2012)"

7. I would recommend including a reference to research that highlights the importance of using scientific evidence to guide decision-making in the context of climate action.
Many thanks, we have added citations up from 2020

Line 60: "Ultimately, the knowledge and strategies for mitigation and adaptation outlined in the IPCC reports are handed over to policymakers, whose decisions impact society at large"

8. It would be interesting to include references to a study of the impact of IPCC reports on policymaking, such as the Paris Agreement.
Many thanks, we have added more references on this matter.

Line 65: "reports form a key part of the information that reaches global society, which must ultimately support the decisions made by political leaders."

9. The order of the words implies the reports should support the decisions of political leaders; do you rather mean, the political leaders should uptake these reports in their decision making?
Many thanks for your point. Our intent is to emphasize that the reports provide a foundation of scientifically grounded information to guide decision-making by political leaders. However, the success of these decisions in addressing climate change also relies on society's collective engagement and action in implementing them.
We have changed the wording to:

These reports provide essential scientific information to guide political leaders in their decision-making and require collective societal action to achieve meaningful progress in addressing climate change.

Line 67: This strategy should ensure that people of all knowledge levels can understand and engage with the content (Doran et al., 2023; Rödder & Pavenstädt, 2023)."

10. To conclude the introduction section, I would suggest adding a sentence stating the aim of this paper, i.e. how this study aims to contribute to such a communication strategy.

Thanks for your comment, although the next section of the paper is "objectives" that are
1.      To understand the perspectives of IPCC scientists on the current state of climate change and their role in efforts to reduce and mitigate its impacts.
2.      To gather opinions from IPCC scientists on how best to communicate the scientific content of IPCC reports to the public.
3.      To collect proposals from IPCC scientists on how to improve the dissemination of this scientific information to society at large.
We have added this sentence:   "This study aims to explore IPCC scientists' views on climate change, communication of reports, and ways to improve public dissemination."

Line 84: "Additionally, this knowledge should be presented in a way that is not only rational and easy to understand but also resonates with people on emotional and spiritual levels across different cultures (Bolisani & Bratianu, 2018)."

11.I would recommend adding more studies on the effect of empathy, psychology (behavioural science), and overall emotional connection for effective climate communication. Perhaps include how on the other hand, there is a negative effect to contend with, such as climate anxiety.
 Many thanks, we have added more references on this matter.

Methodology

The methodology could include more details on the design of the questions, the review process of the questions, and how the open-ended questions were analysed. Below please find specific comments:

Line 100: "The survey was structured into four sections: general information about the scientists, their perception of the current level of knowledge on climate change, their views on the IPCC reports, and opinions on the communication of these reports to society."

12.How were the survey questions designed and by whom? Was a survey expert consulted?

The authors designed the survey questions. This research is part of a broader research about Climate Change communication to the public from an international point of view. We have surveyed other collectives with similar content structures. Yes we did had different experts consultations, and in the cases of this specific paper, as stated in line 112 "The survey questions were reviewed by a scientist who contributed to both the IPCC AR6 and AR5 reports."

Line 103: "open-ended options for questions related to communication"

13.How were open ended answered analysed? There is little mention of the outcome of this section in the Discussion. Could the answers be anonymized and shared in the annex?

Many thanks for your comment. As you see in the questionnaire we asked them to add suggestions on the various topics. We have mentioned those suggestions in the discussions and conclusions.

Line 104: "The survey questions were reviewed by a scientist who contributed to both the IPCC AR6 and AR5 reports. "

14.Was there a testing phase to assess clarity/ambiguity of the questions before the final survey was sent out?

Many thanks for your question, yes, we did try it with two english speaking colleagues

15.How was this one scientist chosen to be the reviewer and what they reviewed (clarity of questions? Appropriateness of language?), or if the had expertise in survey assessment/design. Results and discussion

The scientist, a contributor to IPCC AR5 and AR6 and a prominent figure in climate science and the IPCC, was deemed well-suited to assess the potential reception of the survey by IPCC scientists. Additionally, one of the authors, from the Communication and Information Department, brings expertise in designing and conducting this type of survey.

Line 154-155: "Many respondents see the primary role of the IPCC reports as providing the best possible information to decision-makers, rather than directly to the general public."

16.Is the statement coming from the open-ended questions?

Yes, this fact is mentioned at the end of the paragraph
"Many respondents see the primary role of the IPCC reports as providing the best possible information to decision-makers, rather than directly to the general public. They believe that the public often accesses these reports through other interpretive channels, as expressed in open-ended responses."

Line 170: "students believe climate change is real and primarily human-induced.."

17.I would suggest to exchange the word 'believe' to 'understand' or similar. In my opinion, it doesn't serve the scientific community to use the word 'believe' in the climate discussion.

Many thanks for your comment, but the word "belive" comes from the reference:

"Findings
A strong majority of respondents believe that climate change is real and largely human-induced; a majority express concern about climate change. Yet, students in the sample hold misconceptions about the basic causes and consequences of climate change."
Wachholz, S., Artz, N., & Chene, D. (2014). Warming to the idea: university students' knowledge and attitudes about climate change. International Journal of Sustainability in higher education, 15(2), 128-141.

Lines 173-182:

18. This paragraph describes the results of the survey where social media is rated as a most efficient platform to communicate, with religious sermons in the other extreme. I would respectfully argue that not knowing what the respondents base this assessment on (personal experience with social media? past collaboration with or presentations during a religious sermon?), it is hard to read an opinion on how 'effective' a platform is.

Thank you for your valuable feedback, which has helped us refine our text.

As we mentioned, our aim was to gather the perspectives of IPCC editors of the Summary for Policymakers in the AR5 and AR6 reports. These editors play a key role in shaping the final presentation of the reports to policymakers, following UNFCCC guidelines. While we acknowledge that their assessments of platform effectiveness may be based on subjective perceptions rather than direct experience, we believe it is important to understand their views on how different channels contribute to disseminating this widely published and discussed information. This insight can inform strategies for improving communication effectiveness

Lines 183-185:

19. These statements could be supported by references to studies on the effect of scientific language, values, etc on scientific credibility.

Many thanks, we have added references and changes the wording

Many respondents suggested the possibility of creating a more concise version of the IPCC summary report specifically for the general public. Our survey findings align with the discussions and recommendations from the IPCC's February 2016 Expert Meeting on Communications and their ongoing implementation. The goal is to deepen understanding of the IPCC's communication efforts within the broader context of climate communication and policy. This may also inspire further ideas on how to strengthen the IPCC's communication strategies (Lynn, 2018).

When discussing how to communicate the contents of IPCC reports to the public, the majority (rated 9,2) believe it is appropriate for these reports to be integrated into university curricula and school education (rated 9,2). The strong agreement among our survey respondents aligns with literature indicating that university students believe climate change is real and primarily human-induced, with the majority expressing concern. However, the studies also reveal misconceptions about the fundamental causes and consequences of climate change (Wachholz et al., 2014).

Respondents also emphasized the importance of making the reports fully accessible to everyone via the internet (rated 9). Social networks (rated 9,3), along with media outlets like television (rated 9) and radio (rated 9), were seen as the most suitable platforms for informing the public. The written press was rated slightly lower (rated 8.8), but still viewed as an important channel. Overall, respondents rated highly the effectiveness of these channels for informing the population. The use of new media aligns with studies suggesting that non-elite actors, such as individual bloggers and concerned citizens, are effective climate change advocates. While mainstream media remains the most frequently discussed, new media and science information sources are strong competitors for audience attention (Newman, 2017).

Lines 202-207:

20.You are listing examples of the most common open-ended questions. It is not then clear why there are references to the two listed proposals. Did participants all coincide with these references? Or are you adding supporting references to their opinions?
Many thanks, we have changed the wording

- Short, simple, and easy-to-understand messages, that may help in making IPCC a power communicating tool, that aligns with litterature (Stocker & Plattner, 2016).

- Demonstrating empathy towards individuals and communities by linking climate change to everyday life and focusing on the future of new generations, while staying true to the content of the reports, that also aligns with the most recent published literature (McBeth et al., 2022).

Line 208:

21.You make no prior reference to the geographical representation of the respondents. Including 'Ethiopia' here seems needless, unless their suggestion is particularly relevant coming from that country. I suggest rephrasing it to 'one notable response included…".

Many thanks we have changed the wording

One notable response from a country from the Global South, highlighted the need to improve the training of those responsible for informing the public about climate change.

Conclusion

Line 214-2016:

22.It is not clear to me the relation between high-profile IPCC authors responding the survey and the ability of the average reader to understand the report. Please clarify, if you agree.
As previously mentioned: As we mentioned, our aim was to gather the perspectives of IPCC editors of the Summary for Policymakers in the AR5 and AR6 reports. These editors play a key role in shaping the final presentation of the reports to policymakers, following UNFCCC guidelines. While we acknowledge that their assessments of platform effectiveness may be based on subjective perceptions rather than direct experience, we believe it is important to understand their views on how different channels contribute to disseminating this widely published and discussed information. This insight can inform strategies for improving communication effectiveness

Line 223-224:
"…expressed dissatisfaction with the inclusion of these topics. This highlights the need for a revised communication strategy that addresses these concerns and enhances the impact of the IPCC report content"

23.It is not clear from this sentence if respondents expressed dissatisfaction with having to answer questions on these topics in this survey, or in the inclusion of these topics in the discussion of climate communication in general. Your call to action to address these concerns would suggest the latter, but it is not clear to me from this sentence which it is. You had mentioned in line 217-218 that one of the respondents, lead author of the IPCC, declined to participate in the survey due to the questions of the survey.

All the participants of the survey were either lead authors and chairs and vice chairs of the IPCC AR5 and AR6. In the text of the paper we mention different questions that were more controversial than others between the participants in the survey.
Our paragraph is:
" The disparity in participant behavior in our survey is evident: while some respondents scored highly on questions involving concepts such as journalistic language, religion, or politics, others either refused to participate or expressed dissatisfaction with the inclusion of these topics. This highlights the need for a revised communication strategy that addresses these concerns and enhances the impact of the IPCC report content (Anseel et al., 2010; Bhandari, 2022; Solecki et al., 2024)."

Please allow us to make a longer response. Our aim is to find ways of turning the best climate change science into a good and understandable information that effectively reaches the public and and facilitate the decisions to fight climate change.

From our first reference we wanted to try to find IPCC experts' reaction to the topics we included in our survey.

Anseel, F., Lievens, F., Schollaert, E., & Choragwicka, B. (2010). Response rates in organizational science, 1995–2008: A meta-analytic review and guidelines for survey researchers. Journal of Business and Psychology, 25, 335-349.

"Findings

First, differences in mean response rate were found across respondent types with the lowest response rates reported for executive respondents and the highest for non-working respondents and non-managerial employees. Second, moderator analyses suggested that the effectiveness of response enhancing techniques was dependent on type of respondents. Evidence for differential prediction across respondent type was found for incentives, salience, identification numbers, sponsorship, and administration mode. When controlling for increased use of response enhancing techniques, a small decline in response rates over time was found."

"Implications

Our findings suggest that existing guidelines for designing effective survey research may not always offer the most accurate information available. Survey researchers should be aware that they may obtain lower/higher response rates depending on the respondent type surveyed and that some response enhancing techniques may be less/more effective in specific samples."

Our second reference in this paragraph is from a book
 that highlights the relation from science and politics, and from this to the public and society as a whole.

Bhandari, M. P. (2022). Getting the climate science facts right: the role of the IPCC. River Publishers.

ABSTRACT

Getting the Climate Science Facts Right - discusses climate change science with reference to the Intergovernmental Panel on Climate Change (IPCC). Addressing climate change is the most important public priority of the 21st Century. Unlike many issues, however, this issue is being driven by both science and its interface with politics. The main institution for bridging this division between science and international politics is the IPCC. As such it is the main source of the facts from which climate change policy is developed. This book describes the ways in which the IPCC arrives at these facts and so can be sure they are complete and evidence based.Seldom in history has science had such a direct relationship with politics. The negotiation of an international policy regime requires, at its outset, an agreement on the facts. In this case, the facts are scientific, complex and contentious. Governments have recognized this and have, by using the IPCC, set up institutional machinery to provide facts from a source and in a manner that they can accept.The way in which the IPCC functions is unique in that it melds the way in which science achieves consensus with the way governments do at the international level. Starting with a process to examine, review and debate scientific findings leading to a consensus about scientific fact, usually expressed as probabilities that the findings will hold over time, the IPCC then concludes by using the kind of consensus-development mechanism that the United Nations typically uses to achieve agreements leading to the formation of policy regimes.The book examines the structure of the IPCC, its composition and its procedures in order to achieve an understanding of its role and future.

The third reference emphasizes the need of finding new approaches to deliver this scientific content to the broader populations, for example in the cases of cities and with the purpose of adaptation and mitigation.

Solecki, W., Roberts, D., & Seto, K. C. (2024). Strategies to improve the impact of the IPCC Special Report on Climate Change and Cities. Nature Climate Change, 14(7), 685-691.

Abstract

The planned Special Report on Climate Change and Cities represents a key opportunity to connect the IPCC assessment process to the topics of cities and global urbanization, which are both critical elements of climate adaptation and mitigation during the current 'decade of action'. To help seize this opportunity, we recommend the development of inreach and outreach strategies that can help the report to have greater impact. The new strategies could allow interest groups, including practitioners and policymakers, along with researchers and IPCC representatives to be more coordinated and enhance the utilization of the assessment results. These advances would be useful not only for the upcoming Special Report but also for future IPCC reports and other comparable scientific assessments.

Line 255:

24.You suggest using "clear, accessible language—akin to journalistic style" for the general public; however over 30% of your respondents (figure 3) seem to have "low-" or "very low

agreement" that such an action would make the report easier to understand. Could you please add a comment on this to the discussion?

Many thanks for your opinion. The full text of the paragraph is :
Developing a methodology within the IPCC itself to produce texts written in clear, accessible language—akin to journalistic style (Smith & Higgins, 2020) —could help reduce the contradictory and confusing headlines that often reach the public. Some IPCC scientists who responded to our survey suggested that this could be an innovation for future cycles, proposing ideas such as creating a summary text for the general public, approved by scientists rather than governments, to eliminate concerns about politicization and preserve trust in both the message and its source

Two of the IPCC group of editors suggested creating a summary text for the general public, approved by scientists rather than governments, to eliminate concerns about politicization and preserve trust in both the message and its source. We suggest that a "clear, accessible language" that it is how journalistic texts are written (that is why we add the reference "Smith, A., & Higgins, M. (2020). The language of journalism: A multi-genre perspective. Bloomsbury Publishing USA.", a book that also includes scientific journalism as a powerful tool to communicate science to the public.

Line 267-268:

25.When discussing a gap in effective communication between scientists and the public, I would suggest bringing to the discussion the concepts of knowledge brokers or boundary organisations, which is not discussed at all in this study. For example, a call for knowledge brokers and boundary organisations – albeit in the context of evidence-informed policy making- is the European Commission's JRC "Commission staff Working Document" (SWD(2022) 346).

Many thanks, we absolutely agree with you. In fact, as part of our broader research, we also interviewed participants of all levels of accreditations to the COP 26 in Glasgow, and members of organizations like Extinction rebellion, Fridays for future, Greenpeace,..., and governmental representatives.
In the case of this paper, we are showing the results of the survey with the IPCC Editors

3. Technical corrections

26.Some decimal points are marked by periods (full stops) and some by commas. Please make it consistent. eg. Line 123-126: "The scientists who responded to the survey (figure 1)

were aged 51 and older, with a significant portion (58.4%) over 61 […] The majority of respondents were men (75%) and from academic institutions (83,3%)"

MANY THANKS!! Sorry! we have amended the text

Line 132 -
7.Figures captions don't need to capitalise each word, eg in: "Age, Gender, Profession & Feelings about Climate Change"

Many thanks, we have amended the text

Line 134-135:
"Those with a deeper understanding of the current climate situation tend to view its potential future with greater concern and alarm."

28.I understand you are referring to the surveyed IPCC scientists versus the general public in this line. It may be helpful to clarify you are comparing the respondents to the general public to avoid confusion since the previous line is comparing the different levels of worry of only the surveyed scientists.

Many thanks. We have changed the text, and erased "greater"

Line 159- Figure 3 caption

29. You use the word "Summary" report, and in the figure it is 'Synthesis report'. Please make consistent.
Thanks! we have amended the text

30. The phrase 'summary journalistic language' is not proper English. Please amend.
Many thanks We have amended the figure

Line 189:
31.Figure 4: The figure legend has a typo: "No effective" (green). Change to 'Not effective'.
Many thanks We have amended the figure

Line 216:
32.Order of reference out of chronological order. Supplement The supplement pdf seems to be have truncated questions. Kindly verify the file is correct.

Many thanks, every page of the PDF corresponds to one question, every line is the response for each of the participants, always in the same order.